# Effect of Pre-Corrosion Pits on Residual Fatigue Life for 42CrMo Steel

**DOI:** 10.3390/ma12132130

**Published:** 2019-07-02

**Authors:** Dezheng Liu, Yan Li, Xiangdong Xie, Jing Zhao

**Affiliations:** 1Hubei Key Laboratory of Power System Design and Test for Electrical Vehicle, Hubei University of Arts and Science; Xiangyang 441053, China; 2School of Urban Construction, Yangtze University, Jingzhou 201800, China

**Keywords:** pre-corrosion pits, residual fatigue life, 42CrMo steel, stress intensity factor

## Abstract

The effect of pre-corrosion pits on residual fatigue life for the 42CrMo steel (American grade: AISI 4140) is investigated using the accelerated pre-corrosion specimen in the saline environment. Different pre-corroded times are used for the specimens, and fatigue tests with different loads are then carried out on specimens. The pre-corrosion fatigue life is studied, and the fatigue fracture surfaces are examined by a surface profiler and a scanning electron microscope (SEM) to identify the crack nucleation sites and to determine the size and geometry of corrosion pits. Moreover, the stress intensity factor varying with corrosion pits in different size parameters is analyzed based on finite element (FE) software ABAQUS to derive the regression formula of the stress intensity factor. Subsequently, by integrating the regression formula with the Paris formula, the residual fatigue life is predicted and compared with experimental results, and the relationship of the stress intensity factor, pit depth, and residual fatigue life are given under different corrosion degrees. The fatigue life predicted by the coupled formula agrees well with experiment results. It is observed from the SEM images that higher stress amplitude and longer pre-corroded time can significantly decrease the residual fatigue life of the steel. Additionally, the research work has brought about the discovery that the rate of crack extension accelerates when the crack length increases. The research in this paper also demonstrates that the corrosion pit size can be used as a damage index to assess the residual fatigue life.

## 1. Introduction

Most engineering materials are subject to corrosion, and corrosion research has received much attention from different perspectives [1,2,3,4,5,6,7]. The 42CrMo (American grade: AISI 4140) is a commonly used ultrahigh-strength steel. Due to its high strength, toughness, and hardenability, 42CrMo steel is widely used in quenched and tempered heavy forgings to build components such as pressure vessels, gears, vehicle axles, and deep oil drilling rod subs [8,9]. However, their corrosion resistance is relatively poor and this material is prone to corrosion. Corrosion pits can weaken the strength of the structure and decrease its fatigue life [10,11], since the fatigue cracks are easy to nucleate at corrosion pits and propagate rapidly under dynamic loads [12]. Thus, to further understand corrosion effects on the strength and residual fatigue life of 42CrMo material, the stress state around corrosion pits should be reasonably estimated.

In recent years, fatigue performance of pre-corroded metallic alloys has been studied extensively. For example, recent works [13,14,15,16,17,18] have reported the description of the stress concentration arising from corrosion pits based on the finite element (FE) method. Sharland [13] developed a mechanistic model of the propagation stage of an established pit or crevice to simulate the evolution of pit geometry and to describe the corrosion process. However, Wang [14] indicated that a detailed description of both the chemical reactions and the ionic transport was not included in it [13]. Through the use of interaction function to simulate the effect of the corrosion potential and corrosion current density on the corrosion process, Xu [15] developed an FE model for the simulation of the mechanoelectrochemical effect of pipeline corrosion. Because the fatigue strength of the metallic alloys decreases differently under the different corrosion depths and stress amplitudes [16], Pidaparti [17] established a new FE model to investigate the stress state around corrosion pits. However, the proposed FE model in [17] cannot be used to depict the growth process of corrosion pits with time. Through the use of the damage tolerance method to predict the corrosion fatigue life for 7075-T6 stainless steel, Huang [18] indicted that the nucleation time of pits and the material constant of crack propagation can affect the residual fatigue life of stainless steel significantly. On the basis of [16,17,18], Zheng [19] proposed that the decrease of fatigue life depends on the pre-corroded time and the size of pre-corrosion pits. Most recent studies on the fatigue life of steel in corrosion environments only consider the factors such as mass loss rate and corrosion pit depth, and rarely involve the stress intensity factor varying with corrosion pits in different size parameters. The stress concentration caused by corrosion pits can dramatically influence the basic macroscopic mechanical properties of a steel [20]. Despite the extensive research on the corrosion of metallic alloys [21,22,23,24], there are few detailed studies of the effect of pre-corrosion pits on residual fatigue life for 42CrMo steel.

In this study, the effect of stress distribution around corrosion pits on the stress intensity factor is firstly investigated. The crack nucleation sites and the geometry of corrosion pits are examined by a surface profiler and a scanning electron microscope (SEM), and then the relationship between the fatigue life and the maximum applied stress is obtained by fatigue tests of different pre-corroded specimens. Based on experiment results, an FE model is conducted to investigate stress distribution around corrosion pits, and regression formula of the stress intensity factor varying with corrosion pits in different size parameters is then derived. Subsequently, by coupling the regression formula with the Paris formula [25], the fatigue performance is predicted by the FE method and then compared with experimental tests. Under the different stress amplitudes and pre-corroded times, the FE predictions are finally validated by comparison with the experiment results.

## 2. Materials and Methods

### 2.1. Material and Specimen Preparation

To better understand the stress intensity factor varying with corrosion pits in different size parameters and to accurately depict the residual fatigue life of 42CrMo steel subject to corrosion, it is necessary to quantitatively investigate the effects of pre-corrosion levels on the fatigue behavior of 42CrMo steel. Experimental specimens were fabricated using the 42CrMo steel (American grade: AISI 4140), which was provided by Baowu Steel Company (Wuhan, China). Specimens were manufactured in the form of flat bare sheets with a thickness of 6 mm by wire cutting machine, the sizes of specimens are provided in Figure 1. The chemical composition was measured using an FLS980-stm Edinburgh fluorescence spectrometer (Edinburgh Instruments Ltd., Livingston, UK). A comparison of the measured chemical composition of 42CrMo specimens and the standard of GB/T3077-2015 (the national standard of the People’s Republic of China for alloy structural steel) for 42CrMo steel is presented in Table 1. It can be seen from Table 1 that the chemical composition of the specimens used in this study meets the requirement of the national standard.

Twelve smooth samples were numbered and divided into four groups. The accelerated neutral salt spray corrosion experiment was established to achieve the pre-corroded steel plates based on the standard of JIS H 8502.5 [26]. A schematic drawing of the sample preparation and the pre-corroded experiment procedure is shown in Figure 2. The sample preparation was carried out in five steps: (1) samples were manufactured in the form of flat bare sheets with the dimensions in Figure 1 by wire cutting machine; (2) the surface of each sample was burnished by the 1000 grit abrasive paper to dispel burrs; (3) samples were rinsed in distilled water; (4) samples were degreased through the use of acetone solvent; (5) all surfaces of samples were rinsed with deionized water and dried by a blower for the pre-corroded experiment.

In this study, the pre-corroded experiment procedure can be devised as four main steps: (1) the pure NaCl crystals of 50 g were weighed by an electronic balance and poured into one liter of deionized water and stirred with glass rod at an ambient temperature of (25 ± 1 °C) for 5 min; (2) four groups of numbered samples were alternately immersed in 5% NaCl solution at an ambient temperature of (25 ± 1 °C) for 0 h, 24 h, 48 h, and 96 h, respectively; (3) the corrosion products were removed by membrane removal solution (prepared with 100 mL HCl, 100 mL deionized water and 0.6 g C6H12N4); (4) all surfaces of samples were rinsed with deionized water and dried by a blower for pit depth measurement.

### 2.2. Pre-Corrosion Pit Measurements and Fatigue Test

To achieve the morphology of the corrosion surface and to determine the size and geometry of pre-corrosion pits, a non-contact Dektak150 surface profiler (Veeco Instruments Shanghai Co., Ltd., Shanghai, China) and an S-4800 scanning electron microscope (Hitachi, Tokyo, Japan) were used in this study. A region of 30 mm × 20 mm (along the directions of longitude and transverse, respectively) for each specimen was approximately arranged at the area of pre-estimated fatigue fracture. It is noted that the rate of corrosion changes steadily with the increase of pre-corroded time [27], thus the relationship between the corrosion time and the depth of pre-corroded pits can be expressed by an exponential function. Based on reference [28], the coefficient of exponential function can be determined by fitting the measured data, and exponential function expression can be described as:(1)$$\varphi =1.235\;t^{0.775}$$
where *φ* is the depth of a pre-corroded pit (μm) and *t* is the pre-corroded time (h).

Based on the standard of ASTM E468-90 [29], the fatigue performance of corroded specimens (24 h, 48 h, and 96 h) were evaluated by a hydraulic universal testing machine (Instron-8803, Norwood, MA, USA) under a cyclic load amplitude with a frequency of 10 Hz. The loading operation was force controlled with a proportional error of ±1%. A group of un-corroded specimens were also tested as a control for fatigue studies. The stress ratio was taken as 0.1, and the maximum applied stresses were 100 MPa, 200 MPa, and 300 MPa, respectively. After testing, the fractured specimens were processed by cutting machine and the morphology of the corrosion surface was observed by surface profiler and scanning electron microscope.

The pre-corrosion pit reduced the fatigue life substantially, particularly the interacting pits and sharp pits [30]. In this study, the Paris theory [25] was applied to analyze the relationship between the morphology of fatigue cracks and the stress state around corrosion pits. According to Paris theory [25], the stress intensity factor of cracks can be described as:(2)$$K_{I}=\sigma \;\sqrt{\pi a}f\left(\beta \right)$$
(3)$$\beta =\frac{a}{b}$$
where *K_I_* is the stress intensity factor of the crack, *σ* is the external stress under the plane stress state, and *a* is the crack length and *b* is the specimen width.

### 2.3. FE Model

The FE analysis model of the 42CrMo steel in the form of a corroded flat sheet with a thickness of 6 mm was created using the HyperWorks software (HyperWorks 11.0, Altair Corp., Troy, MI, USA). On the basis of the experiment test, a single semi-elliptical pit model was conducted, and the symmetry plane of the flat sheet and refined mesh around the pit are shown in Figure 3.

The CPS8 quadrilateral element type was adopted in the FE model and cracks were prefabricated on the surface through the use of an assigned seam function that was provided by HyperWorks. The sweeping method [31] was used to generate the mesh in the integral region and the singularity of the mesh was controlled by the mesh regeneration technique [32]. The use of the above techniques [31,32] with local refinement and appropriate local mesh density can improve the precision of the calculation. Subsequently, the FE analysis model with a preset corrosion pit was imported into the ABAQUS software (ABAQUS 6.10, Dassault Systemes Simulia Corp., Johnston, RI, USA) to analyze the stress and fatigue performance of the specimens.

## 3. Results and Discussion

### 3.1. Corrosion Surface Characterization

When the 42CrMo specimens were immersed in NaCl solution, the Cl element in the solution adhered to the surface of specimens and reacted with the Fe^2+^ in the metal to form a soluble clathrate, which resulted in the anodic dissolution of the metal surface and the formation of corrosion pits. With the increase of pre-corroded time, more metal dissolved in the anode and the size of corrosion pit became larger and larger [33]. The specimens pre-corroded for 0 h, 24 h, 48 h, and 96 h were examined by an S-4800 scanning electron microscope and the morphologies of corrosion surface characterization are shown in Figure 4.

According to Figure 4, corrosion surface characterization can be clearly observed and the obvious corrosion pits after pre-corrosion can be found within the crack source area in each diagram. As shown in Figure 4a, there was no corrosion pit on the surface of specimens without pre-corrosion. With the pre-corrosion treatment, Figure 4b shows that there were local micro-pits and pits distributed densely. With the increase of pre-corroded time, Figure 4c,d shows that corrosion pits became larger since the small pits were interconnected to form larger corrosion pits. A series of electrochemical corrosion processes occurred when the specimen was immersed in the NaCl solution. In this study, the corrosion process can be generally divided into two stages: 1) the initiation of the pits stage and 2) the pit development stage. In the first stage, adsorption and agglomeration of chloride ions took place at certain weak sites and the corrosion pits formed. In the second stage, the pits developed as a result of the anodic dissolution of the metal [34,35,36]. Thus, the small pits were interconnected to form larger corrosion pits.

To obtain the effect of the stress distribution around corrosion pits on the stress intensity factor and the residual fatigue life of corroded steel, the size of pre-corroded pits should be accurately assessed and the depth of pre-corroded pits can be measured by a surface profiler. The schematic diagram of pre-corrosion pits under the pre-corroded times 24 h, 48 h, and 96 h is shown in Figure 5. From Figure 5, one can see that the depth of the maximum corrosion pit can be regarded as the radius of a semi-circular surface crack. The depth of the maximum corrosion pit was measured by a surface profiler and an SEM.

### 3.2. Fatigue Test Results and Discussion

Pre-corrosion not only reduces residual fatigue life, but also leads to the change of the stress intensity factor. Under the cyclic load with a frequency of 10 Hz, the local stress concentration around the corrosion pits accelerated the process of fatigue damage and resulted in the reduced fatigue life of the specimen. Three constant stress levels (100 MPa, 200 MPa, and 300 MPa) were carried out to pre-corroded specimens for 0 h, 24 h, 48 h, and 96 h. The fatigue testing results of specimens with different exposure times are shown in Table 2.

Table 2 shows that the fatigue life decreased obviously with the increase of pre-corroded time under the same constant stress level. Furthermore, the fatigue life decreased approximately by more than 12% when the pre-corroded time was doubled, and the fatigue life decreased approximately by more than 11% when the applied stress level was doubled, indicating that both pre-corroded time and applied stress level can significantly affect the fatigue life of specimens, which shows a good agreement with the previous works [37,38]. Although the external load is normally invariant in practical engineering structures and residual fatigue life usually depends on corrosion surface condition and the nominal stress level, the raised nominal stress caused by the reduction of the cross-sectional area of components due to corrosion still decreases the fatigue life of steel [39]. After fatigue testing, the fracture surface of the corroded specimens under 100 MPa was observed by SEM, and the results are shown in Figure 6.

Figure 6a shows the fracture surface of the non-corroded specimen under the constant stress of 100 MPa, and it can be seen that the crack propagated along grain boundaries. Figure 6b–d reveal that the crack originated at the tip of the corrosion pit and propagated along the grain boundaries, and the white highlight area phenomena was caused by the deposition of corrosive elements. Due to the existence of corrosion defects, stress concentration occurred around the sharp intrusion at the pit bottom and accelerated the process of fatigue damage.

### 3.3. FE Investigations

Based on Equations (1)–(3), ABAQUS was applied to compute the stress intensity factor corresponding to different corrosion pit morphology characteristics, which were depicted in the previous section. The same material properties and loading conditions in Section 2.2 were applied to the FE models, and the computed stress distributions around the crack tip are shown in Figure 7.

The ashy area in Figure 7 represents the yielding zone in specimens. The numbers 1, 2, and 3 in Figure 7 denomination represent the constant stress level of 100 MPa, 200 MPa, and 300 MPa, respectively. From Figure 7, it can be seen that, under the same pre-corroded time, the yielding zone increased with the increase of applied loads through the transverse comparison. Under the same applied loads, it can be seen that the yielding zone increased with the increase of the size of pre-corrosion pits. Moreover, the FEM investigations revealed that the rate of crack extension accelerated when the crack length increased.

### 3.4. Regression Formula of the Stress Intensity Factor

By the adoption of Equations (2) and (3), the stress intensity factor around the crack tip in the process of crack propagation for non-corroded specimens can be calculated and compared with the stress intensity factor that was computed by ABAQUS software to validate the accuracy of the FE simulation results. A comparison of the stress intensity factor by theoretical Paris formula and FE simulation is shown in Figure 8.

The maximum percentage error of the calculated stress intensity factor between the theoretical Paris formula and the FEM is 4.7%, which indicates that the theoretical calculation agreed well with the simulated results. Through the integration of the ashy area around the crack tip in Figure 7, the stress intensity factor of the crack tip for pre-corroded specimens can be obtained and the relationship between the stress intensity factor at the crack tip and the depth of the pre-corrosion pits can be established.

Figure 9 shows the relationship between the stress intensity factor at the crack tip and the size of pre-corrosion pits. By using the numerical fitting method, the regression formula to represent the relationship between the stress intensity factor and the depth of the pre-corrosion pits is derived as:(4)$$\frac{K_{IS}}{K_{I0}}=0.273e^{3.378\varphi }+0.731$$
where *K*_IS_ is the pre-corroded stress intensity factor, *K*_I0_ is the non-corroded stress intensity factor, and *φ* is the depth of corrosion pits. The fatigue life can be estimated according to the Paris formula [25]. The Paris formula is shown as follows:(5)$$\frac{da}{dN}=C\Delta K^{m}$$
where *da/dN* is the fatigue crack propagation, Δ*K* is the stress intensity factor at the tip of cracks, and *C* and *m* are the equation coefficients, respectively. For the 42CrMo steel, the coefficients of *C* and *m* are taken as 4.349 × 10^−12^ and 3.07, respectively [40]. Through the integration of Equation (5), the fatigue life of 42CrMo steel can be estimated as:(6)$$N=\int \frac{da}{C{\left(\Delta K\right)}^{3.07}}$$

By combining Equations (4) and (6), the relationship between the fatigue life of pre-corroded and non-corroded specimens can be written as:(7)$$N_{C}={\left(0.273e^{3.378\varphi }+0.731\right)}^{-3.07}N_{0}$$
where *N_C_* is the fatigue life of pre-corroded specimens and *N*_0_ is the fatigue life of non-corroded specimens.

Under different stress ranges and corrosion degrees, the residual fatigue life of pre-corrosion specimens is predicted through the use of Equation (7). The comparison of predicted fatigue life and experimental fatigue life is summarized in Table 3.

Table 2 shows that the fatigue life predicted by the coupled formula agreed well with experiment results, and the percentage errors between predicted values and experimental values were less than 10%. In the case of low corrosion degrees and low stress levels, the percentage errors were less than 5%. With the increase of pre-corroded time and stress levels, the percentage error was increased to approximately 9.3%. The stress intensity factor at the tip of cracks was calculated on the basis of the linear elastic model and the morphology of the pre-corrosion pits simulated by FEM took pre-set defects in the middle of the specimen, while the practical corrosion pits had the characteristics of the uneven distribution and size of the specimen. Thus, the percentage errors between the predicted values and the experimental values were slightly increased with the increase of pre-corroded time and stress levels.

## 4. Recommendation for Future Research

In this research, the effect of pre-corrosion pits on the residual fatigue life of the 42CrMo steel was experimentally and numerically studied. It was demonstrated in this paper that the corrosion pit size can be used as a damage index to assess the residual fatigue life. Therefore, the monitoring of corrosion pit size is important not only to know the corrosion status, but also to predict the residual fatigue life of the steel. With the recent development of structural health monitoring technology [41,42,43,44], especially the piezoceramic-based active sensing method [45,46,47], electromechanical impedance (EMI) approach [48,49,50], and imaging technology [51,52], it is now possible to monitor corrosion pit number and size [10]. Future work should include the real-time monitoring of the onset and growth of the corrosion pit and the prediction of the residual fatigue life of the 42CrMo steel specimens. Future works should also focus on the quantitative analysis of corrosion pits to improve the predicted accuracy of the residual fatigue life for the corroded 42CrMo steel.

## 5. Conclusions

In summary, this study investigated the effect of pre-corrosion pits on the residual fatigue life of the 42CrMo steel. It was found that higher stress amplitude and longer pre-corroded time significantly decreased the residual fatigue life of the steel. The investigation of the residual fatigue life of pre-corrosion specimens was conducted through the use the regression formula and experimental measures, and the fatigue life predicted by the regression formula agreed well with experiment results. Moreover, it was found that the rate of crack extension accelerated when the crack length increased. It is also demonstrated in this paper that the corrosion pit size can be used as a damage variable to assess the residual fatigue life. The recommendation for future research is to monitor corrosion pit characteristics and to predict the residual fatigue life of the 42CrMo steel.

## Figures and Tables

**Figure 1 materials-12-02130-f001:**
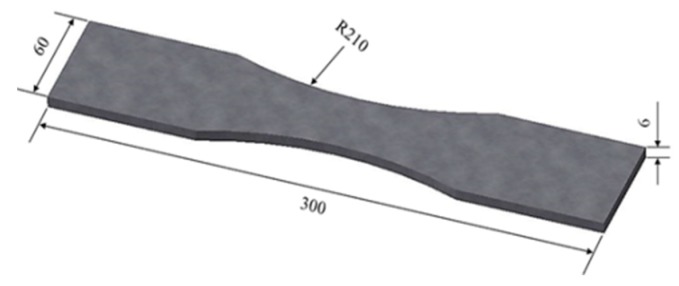
Schematic diagram of specimen sizes (mm).

**Figure 2 materials-12-02130-f002:**
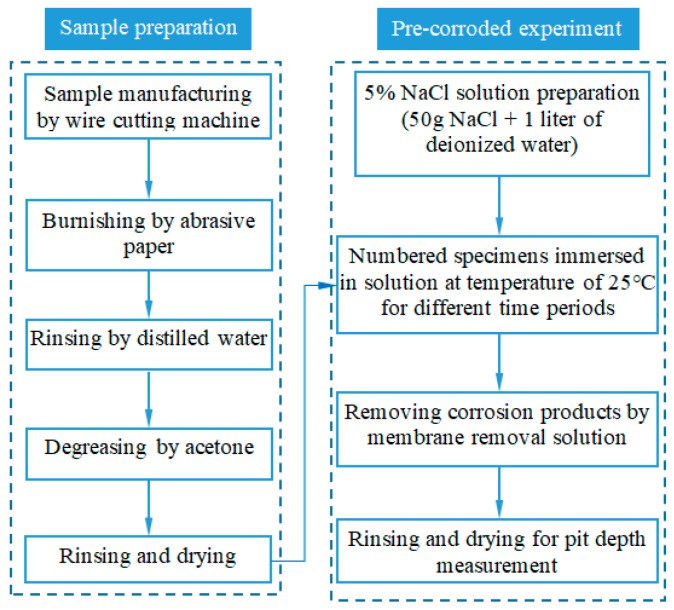
Schematic drawing of the sample preparation and the pre-corroded experiment procedure.

**Figure 3 materials-12-02130-f003:**
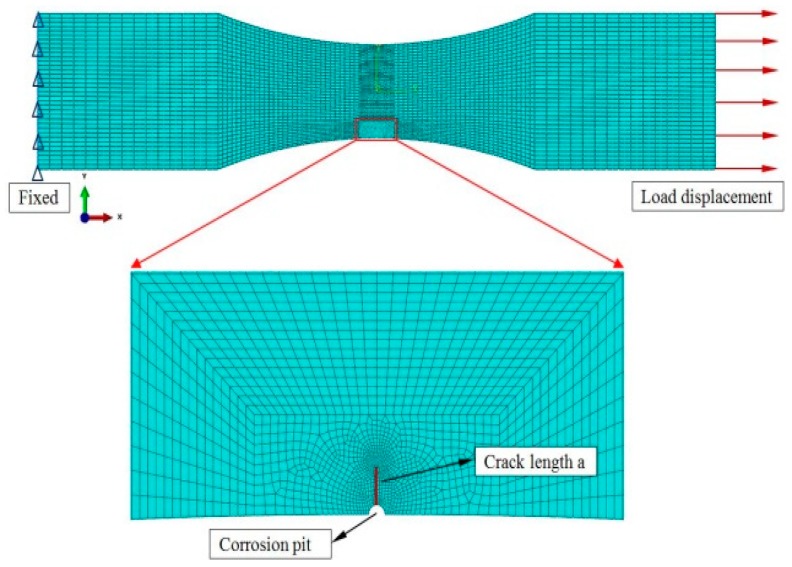
The FE analysis model.

**Figure 4 materials-12-02130-f004:**
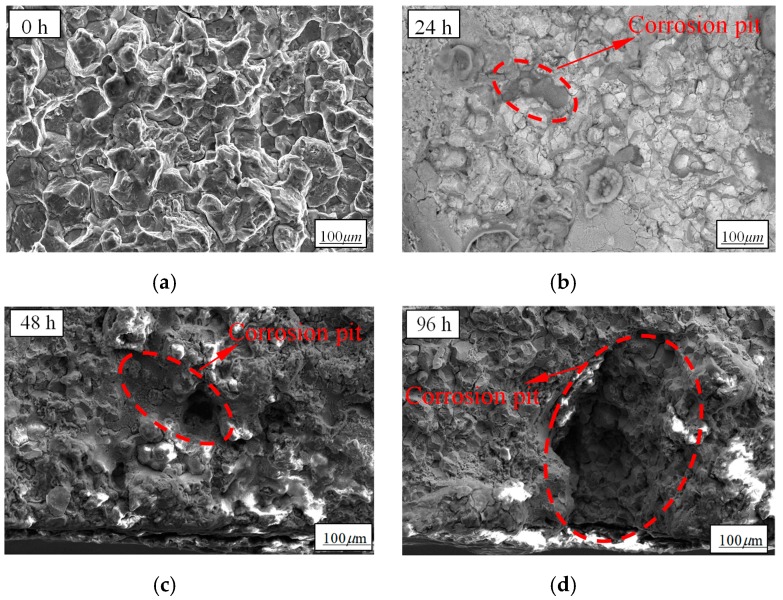
Corrosion surface characterization: (**a**) pre-corroded for 0 h; (**b**) pre-corroded for 24 h; (**c**) pre-corroded for 48 h; and (**d**) pre-corroded for 96 h.

**Figure 5 materials-12-02130-f005:**
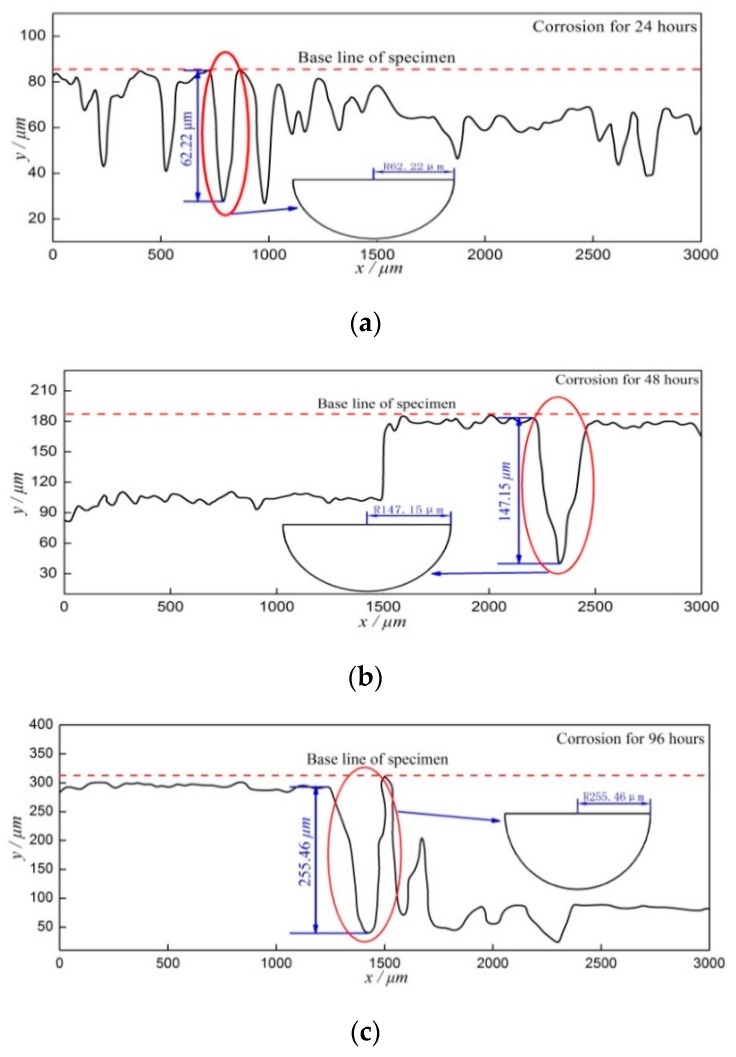
The schematic diagram of pre-corrosion pits under the different pre-corroded times: (**a**) pre-corroded for 24 h; (**b**) pre-corroded for 48 h; and (**c**) pre-corroded for 96 h.

**Figure 6 materials-12-02130-f006:**
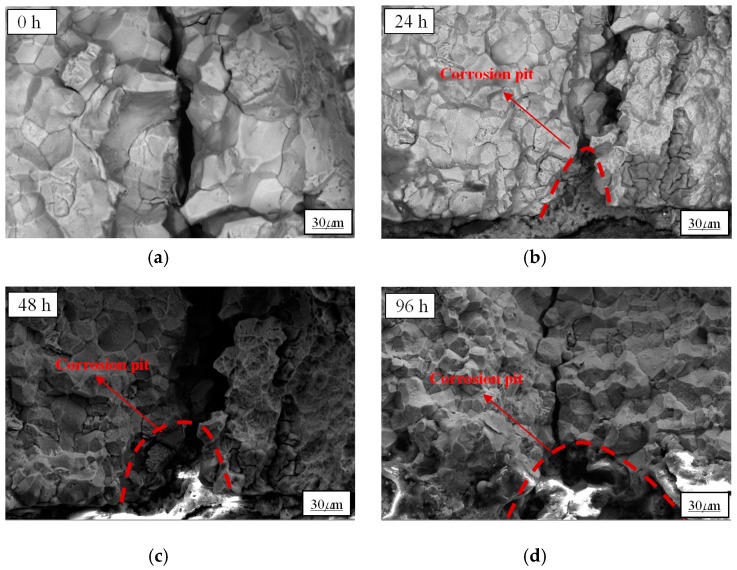
The fracture surface of the corroded specimens under 100 MPa: (**a**) pre-corroded for 0 h; (**b**) pre-corroded for 24 h; (**c**) pre-corroded for 48 h; and (**d**) pre-corroded for 96 h.

**Figure 7 materials-12-02130-f007:**
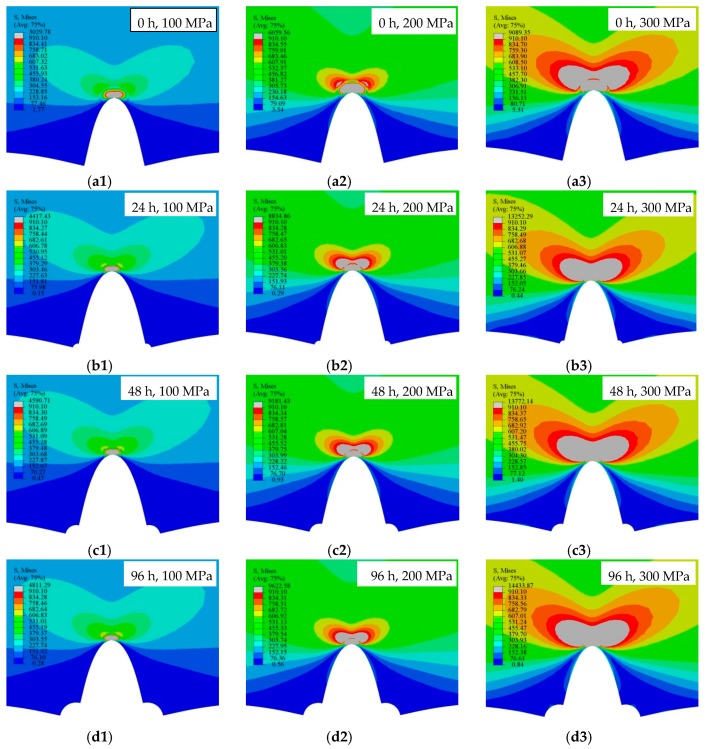
The stress distributions around the crack tip. (**a1**), (**a2**), and (**a3**): non-corroded specimen under the different constant stress levels; (**b1**), (**b2**), and (**b3**): 24 h pre-corroded specimen under the different constant stress levels; (**c1**), (**c2**), and (**c3**): 48 h pre-corroded specimen under the different constant stress levels; and (**d1**), (**d2**), and (**d3**): 96 h pre-corroded specimen under the different constant stress levels.

**Figure 8 materials-12-02130-f008:**
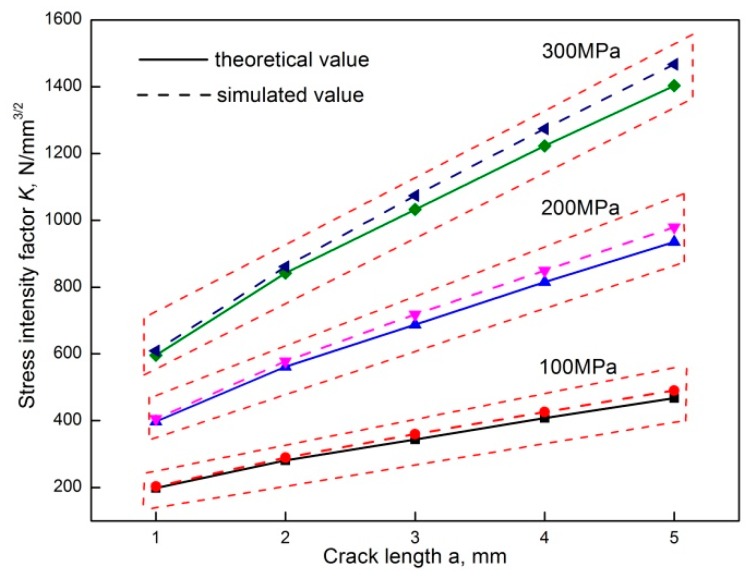
Comparison of the stress intensity factor by theoretical Paris formula and FE simulation.

**Figure 9 materials-12-02130-f009:**
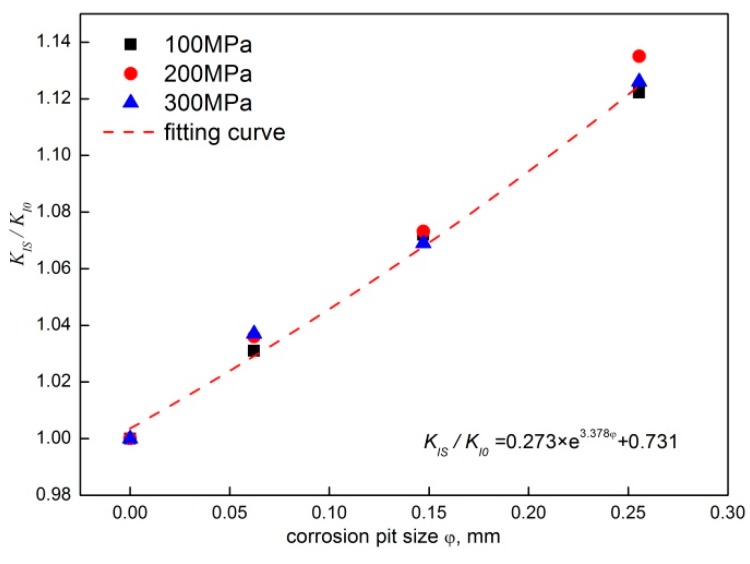
The relationship between the stress intensity factor and the depth of pre-corrosion pits.

**Table 1 materials-12-02130-t001:** A comparison of the measured chemical composition of 42CrMo specimens and the standard of GB/T3077-2015 for 42CrMo steel (wt.%).

wt.%	C	S	P	Cr	Ni	Mn	Si	Cu	Mo
42CrMo	0.42	0.002	0.022	1.08	0.02	0.69	0.28	0.02	0.18
GB/T3077-2015	0.38–0.45	≤0.035		0.90–1.20	≤0.30	0.50–0.80	0.17–0.37	≤0.25	0.15–0.25

**Table 2 materials-12-02130-t002:** The fatigue testing results of specimens with different exposure times.

Pre-Corroded Time (h)	Fatigue Life (×10^5^ Cycles)		
	100 MPa	200 MPa	300 MPa
0	11.26	9.95	7.29
24	9.81	7.91	5.47
48	7.52	6.21	4.09
96	5.03	3.91	2.61

**Table 3 materials-12-02130-t003:** Comparison of predicted fatigue life and experimental fatigue life.

Corrosion Pit Size (μm)	Load (Stress/MPa)	Experimental Values (×10^5^ Cycles)	Predicted Values (×10^5^ Cycles)	Errors (%)
62	100	9.82	10.13	3.15
	200	7.53	7.85	4.08
	300	5.02	5.31	5.46
147	100	8.24	8.73	5.61
	200	6.32	6.75	6.37
	300	4.21	4.53	7.06
255	100	5.62	6.61	8.17
	200	4.43	5.24	8.85
	300	2.95	3.25	9.23

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
