# Peer review of "Effect of Pre-Corrosion Pits on Residual Fatigue Life for 42CrMo Steel"

_materials, 2019, doi:10.3390/ma12132130_

Round 1
Reviewer 1 Report
Does pitting corrosion actually occur on the steel examined by the authors? Looking at the table with the composition of steel, there is no indication of the formation of a passive layer on the surface, the presence of which is a prerequisite for the pitting corrosion.
In this case, should not one talk about general uneven corrosion?
I think that this should be verified in the description.
It is possible to carry out a cyclic polarization test that will clearly show whether the tested alloy is subject to local corrosion or general corrosion.
It probably does not change the further reasoning process because the pitting and the corrosive place faster than the environment will act as a notch.
linie 181 "FeCl" - rather FeCl2
Author Response
Dear reviewer,
We are very grateful to your comments for the manuscript. We also highly appreciate your carefulness, conscientious, and the broad knowledge on the relevant research fields, since you have given us a number of beneficial suggestions. According to the comments we received, we have made the following revisions, which improve the quality of this manuscript.
A revised manuscript with the correction sections red marked (with Track changes) was attached as the supplemental material and for easy check or editing purpose. Thank you very much for giving me an opportunity to revise our manuscript. We hope that these revisions are satisfactory and that the revised version will be acceptable for publication in MATERIALS. I hope you are satisfied with the revised version, however, if there is more question, we are willing to make further revision.
Wish you all the best!
Sincerely yours,
Dezheng Liu on behalf of all authors

Reviewer 2 Report
The manuscript deals with fatigue corrosion of a specific grade of high strength steel. It studies the effect of pits developed during a pre-corrosion phase on the residual fatigue life. Clear effect of the pit depth on reduction of the fatigue resistance is documented. The paper is mostly well written, results are well presented and conclusions supported by sufficient discussion. I recommend the paper for publication in MATERIALS. However, some minor revisions should be carried out before the paper publication.
The paragraphs in chapter 2 starting by line 100 and 110 are badly written. The sample preparation needs to be better explained. What does it mean “rinsed by cotton balls dipped in acetone”? How the samples were pre-corroded? In the neutral salt spray test conditions (standard JIS H 8502.5, which is roughly equivalent to ISO 9227, i.e. 35 oC, salt fog, 5% NaCl), or in 6% NaCl solution at laboratory temperature? Were corrosion products removed before pit depth measurement? How? Maybe a sample photograph would help here. I do not understand the part on profiling as well, lines 114-117. Please, try to explain the procedure in a better way. Schematic drawing(s) could be useful.
Line 74: There is at typo. Instead of Paris, it should read Pairs.
Line 91: Please, explain how the specimens were prepared. By laser cutting, water jet cutting, machining?
Table 1: The second line is explained neither in caption nor text body.
Figures 2, 3 and 4 are unnecessary. These are standard analytical instruments. The denomination is sufficient.
Formula at line 121: Add units.
Line 142, Figure 5: Abbreviations FE and FEA are used in parallel. Please, stick to one of them.
Line 152-153: „The use of above techniques tolerates irregular…”. What does it mean? Please, explain or re-formulate.
Figure 6b: I do not see any pit there. Can you mark it with an arrow?
Lin 175-183: Please, check the explanation of the pitting corrosion mechanism. It is unconventional. There is a lot of literature on this subject. Reference to a well-accepted source should be added.
Line 209: The table does not show that “the fatigue life decreases exponentially”. It may be true but it is not visible in the table.
Line 210-211: “…the fatigue life decreases approximately by more than 12.3%...”. If an approximate is given, I propose to round the number to 12 %.
Figure 9: Image denomination (a)-1, (a)-2 etc. is not explained. I suppose 1 stands for 100 MPa, but it should be written somewhere.
Line 319-320: I do not understand the sentence „Moreover, it was found the effect of pit depth on stress intensity factor at the surface point is higher than the deepest location.“ Please, re-formulate or explain it.
The manuscript should be proof-read again to improve the English. There are clearly incorrect sentences. For example, the sentence on line 49 should read “Because of the fatigue strength …”, line 52 “However, the proposed FE …”, line 57, 162 “… pre-corrosion duration …”, line 91 “Specimens were flat sheets …”, line 159 and elsewhere, use “immersed” instead of “soaked”, Figure 6 and line 168: “the morphologies of corrosion surface characterization” is nonsense, line 215 “residual fatigue life usually depended”, line 303 ”… electromechanical impedance …“, line 304 ”… possible to monitor corrosion …”, line 315 “… and longer pre-corrosion time …”, and others.
Author Response

(The authors gave the same response as above.)

Reviewer 3 Report
Well prepared paper. Only several debatable points need to be reconsidered.
- Line 31-32: Research of reference 2 (Peng et al.) is not focused on corrosion. Thus, the usage These sentences are not the conclusions of the authors but just the procedure information.
- Line 64-77: This part of the introduction is more suitable for Materials and Methods section.
- Line 77-80: This part of the introduction is more suitable for Conclusions section.
- Line 81-87: This part summarizing the content of the paper is useless in the introduction.
- Photos of used experimental devices are unnecessary (Figures 2, 3 and 4).
- Line 173: “…larger…” instead of “…lager…”
- Line 174-183: I do not suppose that this theory is the original theory of the authors. Hence, it need to be cited.
- Line 303: “…electromechanical…”
- Line 310-315: The 2nd and the 3rd sentences of the Conclusions section are unnecessary. These sentences are not the conclusions of the authors but just the procedure information.
Author Response

(The authors gave the same response as above.)
